# The Cytotoxic Effect of Newly Synthesized Ferrocenes against Cervical Carcinoma Cells Alone and in Combination with Radiotherapy

**Hana Skoupilova [1], Vladimir Rak [2], Jiri Pinkas [3], Jindrich Karban [4] and Roman Hrstka [1,*]**

[1] Research Centre for Applied Molecular Oncology, Masaryk Memorial Cancer Institute, Zluty kopec 7, 656 53 Brno, Czech Republic; hana.skoupilova@med.muni.cz

[2] Department of Radiation Oncology, Masaryk Memorial Cancer Institute, Zluty kopec 7, 656 53 Brno, Czech Republic; vladimir.rak@mou.cz

[3] J. Heyrovský Institute of Physical Chemistry, Academy of Sciences of the Czech Republic, v. v. i., Dolejškova 2155/3, 182 23 Prague 8, Czech Republic; jiri.pinkas@jh-inst.cas.cz

[4] Institute of Chemical Process Fundamentals, Academy of Sciences of the Czech Republic, v. v. i., Rozvojová 135, 165 02 Prague 6, Czech Republic; karban@icpf.cas.cz

\* Correspondence: hrstka@mou.cz; Tel.: +420-543-133-306

**Abstract:** Cervical cancer is one of the most common types of cancer in women, with approximately 500,000 new cases and 250,000 deaths every year. Radiotherapy combined with chemotherapy represents the treatment of choice for advanced cervical carcinomas. The role of the chemotherapy is to increase the sensitivity of the cancer cells to irradiation. Cisplatin, the most commonly used drug for this purpose, has its limitations. Thus, we used a family of ferrocene derivatives (in addition, one new species was prepared using standard Schlenk techniques) and studied their effects on cervical cancer cells alone and in combination with irradiation. We applied colorimetric assay to determine the cytotoxicity of the compounds; flow cytometry to analyze the production of reactive oxygen species (ROS), cell cycle, and mitochondrial membrane potential (MMP); immunochemistry to study protein expression; and colony forming assay to evaluate changes in radiosensitivity. Treatment with ferrocenes exhibited significant cytotoxicity against cervical cancer cells, associated with increasing ROS production and MMP changes, suggesting the induction of apoptosis. The combined activity of ferrocenes and ionizing radiation highlighted ferrocenes as potential radiosensitizing drugs, while their higher single-agent toxicity in comparison with routinely used cisplatin could also be promising. Our results demonstrate antitumor activity of several tested ferrocenes both alone and in combination with radiotherapy.

**Keywords:** ferrocenes; chemotherapy; cytotoxic effect; radiotherapy; radiosensitization; irradiation; cell death

## 1. Introduction

Worldwide, cervical carcinoma is the second most common malignancy specific to women. It was estimated that in 2012, more than half a million women were diagnosed with cervical cancer around the world, and approximately quarter of a million died [1,2].

Fortunately, the incidence of cervical cancer is steadily declining in most developed countries [3]. Two main reasons for this trend are effective screening and vaccination against the most common oncogenic human papilloma virus (HPV) strains which cause almost all cervical cancers [4]. Treatment of cervical cancer depends on the stage of disease and ranges from conization (simple removal of abnormal cervical epithelium) or trachelectomy (removal of the whole cervix) to hysterectomy and/or

radiotherapy [5]. Radiotherapy is usually used in more advanced cases, either to remove remaining microscopic disease after surgery (adjuvant therapy) or as a main treatment when surgery cannot be performed (curative therapy).

Curative radiotherapy is often combined with chemotherapy to improve the treatment outcome [6]. The most commonly used chemotherapeutic drug in this case is cisplatin. Adding a weekly infusion of cisplatin to radiotherapy improves five-year overall survival by 6% and disease-free survival by 8% [6].

Although platinum substances are widely used, they can cause serious side effects, including renal, neural, and gastrointestinal toxicity, that limit their usefulness [7]. Thus, not only the advantages but also the disadvantages of cisplatin stimulated further research into other types of compounds containing metal in their structure. Besides platinum complexes, species containing iron, titanium, ruthenium, gold, or palladium have been synthesized and tested [8–14]. Among the most intensively studied substances exhibiting radiosensitization effects are ruthenium compounds [15–17] or gold nanoparticles [16], as demonstrated also by in vivo screenings [18–20]. Nevertheless, despite a great deal of research into iron-containing antitumor compounds, there have been only a few investigations into the potential radiosensitizing effects of these compounds in cancer cells [21,22].

This article is focused on the series of ferrocene derivatives of the general formula [Fe($\eta^5$-C$_5$H$_4$CH$_2$(p-C$_6$H$_4$)CH$_2$(N-het))$_2$] bearing either substituted or unsubstituted saturated five- and six-membered nitrogen-containing heterocycles (Figure 1). In vitro cytotoxicity analysis was performed on cell lines derived from cervical cancer. In the case of highly toxic ferrocenes, we focused especially on their mechanisms of action in terms of the disruption of cell metabolism and examination of cellular mechanisms leading to cell death. The combined effect of ferrocenes with ionizing radiation was also determined to test the possible use of these ferrocenes as potential radiosensitizing agents.

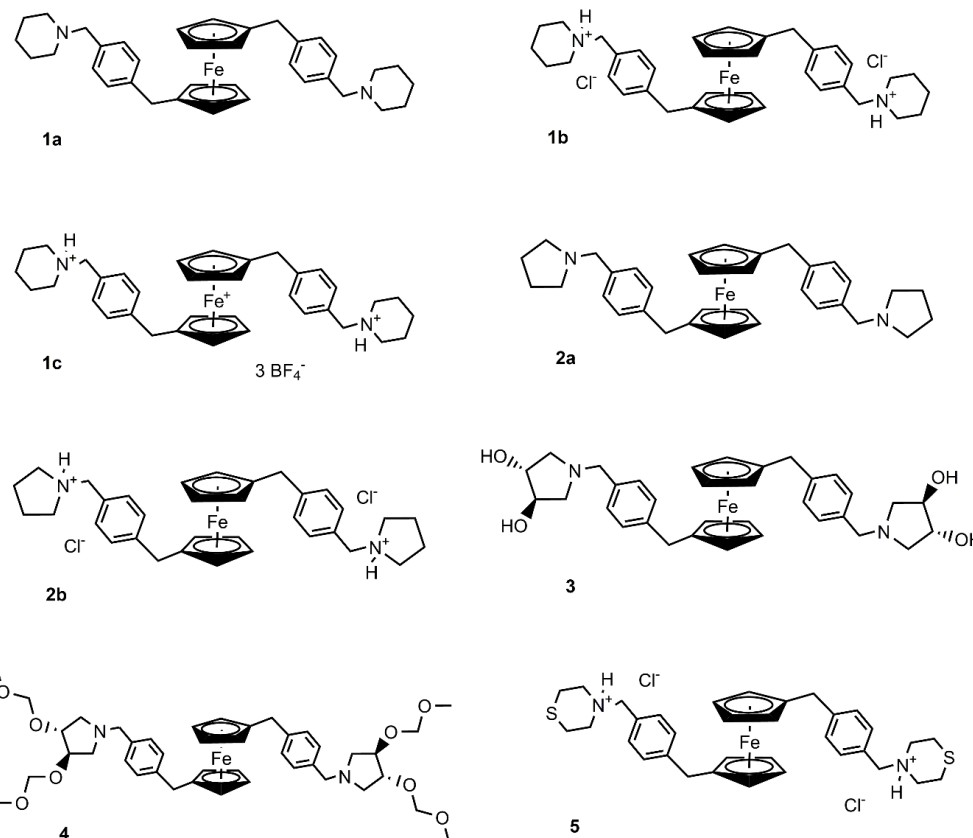

**Figure 1.** Structures of the tested ferrocenes.

## 2. Materials and Methods

### 2.1. Preparation and Characterization of Ferrocenes

Ferrocenes **1a**, **1b**, **2a**, **2b**, **3**, **4**, and **5** were prepared under argon atmosphere using standard Schlenk techniques, as was previously published [23]. Ferrocene **1c** was prepared from **1b** (155 mg, 244 μmol) and AgBF$_4$ (149 mg, 764 μmol) (Scheme 1). The substances were dried under vacuum for 1 h. The vacuum was replaced with argon and dry acetone (10 mL) was added, which caused an immediate color change from yellow to dark green. The mixture was then stirred for 2 days with light exclusion and then filtered. The filtrate was evaporated to dryness to obtain **1c** as a green solid substance with yield 200 mg (99%). M.p. 82 °C. ATR (Si); cm$^{-1}$: 3151 (sh, vw), 3115 (w), 2961 (w), 2932 (sh, vw), 2870 (vw), 1614 (vw), 1518 (w), 1462 (m), 1417 (m), 1367 (w), 1284 (w), 1226 (vw), 1054 (vs), 1035 (sh, s), 857 (m), 839 (sh, w), 764 (vw), 578 (w), 520 (m), 416 (w). Elemental analysis for **1c**—C$_{36}$H$_{46}$B$_3$F$_{12}$FeN$_2$ calculated C, 52.53; H, 5.63; N, 3.40%, found C, 51.41; H, 5.49; N, 3.35%. Its infrared spectrum (ATR) is characterized by the presence of a very strong and broad band at 1054 cm$^{-1}$, which could be assigned to stretching B–F vibration of the [BF$_4$]$^-$ anion.

**Scheme 1.** Preparation of **1c**.

Ferrocene derivatives **1a**, **1b**, **2a**, **2b**, **3**, **4**, and **5** were characterized by elemental analysis, melting point, nuclear magnetic resonance (NMR), electrospray ionization mass spectrometry (ESI-MS), and X-ray diffraction analysis, as reported previously [23,24].

### 2.2. Cell Lines and Cultivation

Cervical cancer cell lines Ca Ski (ATCC® CRL-1550™), SiHa (ATCC® HTB-35™), and HeLa (ATCC® CCL-2™) were used. As nonmalignant controls we used the hTERT (human telomerase reverse transcriptase) immortalized retinal epithelium cell line RPE-1 (ATCC® CRL-4000™) and immortalized HEK 293 cells (ATCC® CRL-1573™) derived from human embryonic renal epithelium. All these cell lines were obtained from the American Type Culture Collection (ATCC, Manassas, Virginia, USA). Ca Ski and SiHa were maintained in high-glucose RPMI-1640 Medium (Sigma-Aldrich, St. Louis, USA) at 37 °C in a humidified atmosphere with 5% CO$_2$. HeLa and noncancerous cell lines HEK 293 and RPE-1 were maintained in high-glucose Dulbecco's modified Eagle's medium (DMEM) (Sigma-Aldrich, St. Louis, Missouri, USA) under the same conditions. Both media were supplemented with 10% fetal bovine serum (Gibco, Thermo Fisher Scientific, Waltham, Massachusetts, USA), 300 μg/mL L-glutamine (Sigma-Aldrich, St. Louis, Missouri, USA), and 100 μg/mL HyClone Penicillin–Streptomycin 100× solution (BioSera, Nuaille, France). The culture medium was changed during each cell passage. Cells were grown to 60–80% confluence prior to experimental treatments with ferrocenes at concentrations from 1 to 100 μM. Cells were mycoplasma-free throughout the duration of all experiments.

### 2.3. Cell Viability Assay

Due to their different sizes and growth rates, SiHa and Ca Ski cells were seeded at a density of 10,000 cells; HeLa, 8000 cells; HEK 293, 5000 cells; and RPE-1, 4000 cells per well in 96-well plates. The next day, the cells were exposed to the tested ferrocenes diluted in dimethyl sulfoxide (DMSO; Sigma-Aldrich, St. Louis, Missouri, USA) in concentrations from 0 to 100 μM (each in pentaplicate) for 24 h. Cell viability was measured using colorimetric MTT (3-(4,5-dimethylthiazol-2-yl)-2,5-diphenyltetrazolium bromide) assay as described previously [25]. All experiments were performed in three independent runs (twice,

when cytotoxicity was over 100 μM). Data from cytotoxicity assays were measured using a Microplate Reader: Infinite® M1000 PRO (Tecan, Männedorf, Zürich, Switzerland) and analyzed using GraphPad Software (San Diego, California, USA) as $IC_{50}$ values (concentrations of compounds that cause metabolic inhibition of 50% of cells).

### 2.4. Cell Cycle

The cell cycle was measured by a modified propidium iodide (PI) staining protocol as described previously [26,27]. Cells in 6-well plates were treated with 5 μM ferrocenes or cisplatin (positive control), and half of the wells were irradiated with 4 Gy (discussed in detail in Section 2.11); all samples were then incubated for 24 h. Afterwards, cells were trypsinized, washed with PBS (phosphate-buffered saline), and centrifuged at 1000 rpm for 5 min. Pellets were washed and resuspended in 0.5 mL of PBS. Cells were then fixed in 70% EtOH for at least 4 h at 4 °C. After fixation, the cells were centrifuged at 1000 rpm for 5 min and washed in PBS again. Subsequently, the cells were stained in 1 mL of staining solution: 0.1% Triton X-100, 10 μg/mL PI (both Sigma-Aldrich, St. Louis, USA), and 100 μg/mL DNase-free RNase A (Invitrogen, Carlsbad, California, USA) for 10 min at 37 °C. The DNA content was measured using a flow cytometer (Navios, Beckman Coulter, USA).

### 2.5. Reactive Oxygen Species (ROS) Production

SiHa or HeLa cells were seeded at 8000 cells per well in dark 96-well plates and incubated under standard conditions for 24 h. The next day, the medium was changed with 100 μL of Hanks' balanced salt solution (HBSS) (0.137 M NaCl, 5.4 mM KCl, 0.25 mM $Na_2HPO_4$, 0.1 g glucose, 0.44 mM $KH_2PO_4$, 1.3 mM $CaCl_2$, 1.0 mM $MgSO_4$, 4.2 mM $NaHCO_3$). Cells were incubated for 1 h. The solution was subsequently aspirated, and 100 μL of HBSS with 5 μM concentration of general oxidative stress indicator 5-(and-6)-chloromethyl-2′,7′-dichlorodihydrofluorescein diacetate, acetyl ester (CM-$H_2$DCFDA; Invitrogen, Carlsbad, California, USA) was added. After 30 min the cells were washed twice and treated with 10 μM ferrocenes or 50 μM $H_2O_2$, serving as a positive control, or 10 mM N-acetylcysteine (NAC), serving as a negative control. ROS production was measured after 2, 4, and 6 h.

### 2.6. Mitochondrial Membrane Potential Changes

The changes in mitochondrial membrane potential were measured using a 1,1′,3,3′-Tetraethyl-5,5′,6,6′-tetrachloroimidacarbocyanine iodide dye—mitochondrial membrane potential probe (JC-1; Invitrogen, Carlsbad, California, USA) [28,29]. The cells were harvested with trypsin and seeded at a density of 0.15–0.2 million of cells per well in 12-well plates, then allowed to adhere overnight. The cells were exposed to ferrocenes in 5 μM concentration for 24 h. Since SiHa cells were more resistant, the conditions were modified to 40 μM concentration for 6 h. Treatment with valinomycin (Molecular probes, Eugene, Oregon, USA) in 50 μM concentration for 2 h served as a positive control. Cells were collected with a rubber scraper, washed with PBS, and centrifuged at 1000 rpm for 5 min. Pellets were washed, centrifuged, and resuspended in PBS with JC-1 probe in concentration 5 μg/mL. Mitochondrial potential changes were measured using a BD FACS Aria sorter (BD Biosciences, Franklin Lakes, New Jersey, USA).

### 2.7. Annexin V–Fluorescein Isothiocyanate (FITC)/PI Binding Assay

HeLa cells were incubated for 24 h at 37 °C in 6-well plates with 10 μM ferrocenes. SiHa cells were more resistant; thus, the conditions were modified to 20 μM concentrations. Cells were harvested by acutase, washed twice with PBS, centrifuged at 1000 rpm for 5 min, and then resuspended in Annexin V Binding buffer (10 mM HEPES/NaOH, pH 7.4; 14 mM NaCl; 2.5 mM $CaCl_2$) at a concentration of 1 million cells/mL. A volume of 100 μL of cell suspension was pipetted into the 1.5 mL tube and mixed with fluorescein isothiocyanate (FITC)-labeled Annexin V (BioLegend, San Diego, USA) and PI solutions. The cells were gently vortexed and incubated for 15 min at 22 °C in the dark. After incubation,

400 µL of Annexin V Binding buffer was added and samples were measured using a flow cytometer (Navios, Beckman Coulter, USA).

### 2.8. Western Blot Analysis

Hela and SiHa cell lines were treated with 2 µM concentration of selected ferrocenes, and half of them were subsequently irradiated with 4 Gy of ionizing radiation. All samples were then incubated for 24 h. Cells were washed twice with ice-cold PBS, scraped off with a rubber scraper, and then lysed in nonyl phenoxypolyethoxylethanol (NP-40) lysis buffer (150 mM NaCl, 50 mM TrisHCl pH 8.0, 5 mM NaF, 5 mM EDTA, 1% NP-40, 1:100 phosphatase inhibitor cocktail, and 1:100 protease inhibitor cocktail, both cocktails from Sigma-Aldrich, St. Louis, USA). The protein concentrations were measured via Bradford protein assay (Bio-Rad, Hercules, California, USA). A quantity of 20 µg of protein lysate per sample was applied and separated on 10% SDS polyacrylamide gel, then transferred onto nitrocellulose blotting membrane (Pall Life Sciences, New York, USA). The accuracy of sample loading was verified with Ponceau staining. Membranes were blocked in 5% milk with 0.1% Tween 20 in PBS and probed overnight with the following antibodies: anti β-actin monoclonal antibody (Sigma-Aldrich, St. Louis, Missouri, USA) served as a loading control, SQSTM1 p62 antibody (A-6) (Santa Cruz Biotechnology, Dallas, Texas, USA), and LC3B antibody (Novus Biologicals, Littleton, USA). Membranes were washed with PBS containing 0.1% Tween and incubated with secondary IgG antibodies SWAR-Px (Swine Anti-Rabbit Immunoglobulins- horseradish peroxidase, #P0217) and RAM-Px (Rabbit Anti-Mouse Immunoglobulins- horseradish peroxidase, #P0161) (Dako, Glostrup, Denmark) for 1 h. Positive signals were visualized with enhanced chemiluminescence reagent (ECL; Amersham Pharmacia Biotech, UK) using a G:BOX Chemi XX6 System (Syngene, Cambridge, United Kingdom). Ordinarily used chemicals were obtained from Sigma-Aldrich (St. Louis, Missouri, USA).

### 2.9. Immunofluorescence Staining

Cells were seeded on coverslips in 12-well plates. After 24 h of incubation at 37 °C with 5 µM concentration of ferrocenes **1b**, **2a**, and **3** and with 0.2 µM Bafilomycin A as a positive control, coverslips with adherent cells were washed with PBS solution and fixed with 4% formaldehyde. After permeabilization with 0.2% Triton-X100 (Sigma-Aldrich, St. Louis, Missouri, USA), cells were incubated with primary antibodies recognizing p62 and LC3B, respectively, for 1 h at 37 °C. The cells were then washed and incubated with fluorescent-dye-conjugated secondary antibodies (ab96899 for LC3B and ab96881 for p62; Abcam, Cambridge, United Kingdom) for 1 h at room temperature. In parallel, Hoechst staining was used to visualize nuclei. PBS was used for washing and Vectashield (Cole-Parmer, Vernon Hills, Illinois, USA) for mounting the coverslip. Cells were visualized on a Nuance Multispectral Tissue Imaging System FX (PerkinElmer, Waltham, Massachusetts, USA).

### 2.10. Colony Forming Assay (CFA)

The radiosensitizing properties of the selected compounds were tested in vitro according to the standardized protocol for colony forming assay [30]. Briefly, cells were trypsinized, centrifuged, and resuspended in fresh medium. The suspensions were then plated at a density of 250 cells per well in 12-well plates and left for 24 h in an incubator to adhere. The next day, selected ferrocenes were added in concentrations of 0.5 and 1.0 µM for HeLa cells and 1.0 and 2.0 µM for the more-resistant SiHa cells. Following 1–2 h of incubation, the cells were irradiated according to the protocol in Section 2.11. The cells were then further incubated under standard conditions for 14 days, and colonies were fixed with crystal violet staining/fixing solution—1% methanol (Penta, Chrudim, Czech Republic), 0.05% crystal violet (Merck Millipore, Burlington, Massachusetts, USA), and 3.7% formaldehyde (Sigma-Aldrich, St. Louis, USA) in PBS and manually counted. Surviving fractions (SF) were calculated by comparing the number of colonies in irradiated and nonirradiated (control) plates using a linear-quadratic model for cell death after irradiation. To obtain the final radiosensitizing effect, dose-modifying factors (DMF)

were calculated as the ratios of surviving fractions in cells irradiated with and without tested ferrocenes at selected doses.

### 2.11. Ionizing Radiation

Orthovoltage X-ray irradiation was performed using an Xstrahl 200 radiotherapy system (Xstrahl, Surrey, England) as a single fraction of 2, 4, or 6 Gy with energy 200 kV with a half-value layer of 1 mm of Cu. The field size was $20 \times 20$ cm, the distance from source to irradiated wells was 50 cm, and the dose rate was 0.38 Gy/min. To obtain an adequate dose at a surface, wells were covered with a 1 cm thick bolus material. Control cells were taken from the incubator and handled similarly to treated cells, except during irradiation when they were placed outside the treatment room to avoid any exposure to ionizing radiation. All treatments were done at room temperature.

## 3. Results

### 3.1. Cytotoxic Activity

The cytotoxicity of ferrocenes was tested in vitro against cervical cancer cell lines CaSki, SiHa, and HeLa using MTT tests. Since nephrotoxicity remains the main side effect of cisplatin treatment, the toxicity of the selected (most active) complexes against noncancerous human embryonic kidney cells (HEK 293) was evaluated along with retinal epithelial cells (RPE-1). Table 1 shows the $IC_{50}$ values for particular ferrocenes. According to the cytotoxicity results, the three most active substances—**1b**, **2a**, and **3**—were selected and used for further testing.

**Table 1.** Cytotoxic effects of the studied ferrocenes in micromolar concentrations after 24 h against selected cell lines.

| Comp. | Ca Ski | SiHa | HeLa | RPE-1 | HEK 293 |
|-------|--------|------|------|-------|---------|
| Cisplatin | $35.1 \pm 8.7$ | $26.6 \pm 4.1$ | $28.2 \pm 5.9$ | $46.8 \pm 6.0$ | $25.3 \pm 8.6$ |
| 1a | $69.4 \pm 8.6$ | $17.1 \pm 3.4$ | $61.5 \pm 2.9$ | $82.8 \pm 3.7$ | >100 |
| 1b | $6.2 \pm 2.1$ | $9.9 \pm 2.2$ | $6.4 \pm 1.6$ | $5.1 \pm 0.8$ | $35.6 \pm 6.2$ |
| 1c | $12.5 \pm 0.6$ | $18.2 \pm 3.1$ | $10.6 \pm 2.5$ | $11.0 \pm 2.4$ | $16.7 \pm 2.0$ |
| 2a | $8.3 \pm 2.3$ | $14.1 \pm 3.4$ | $7.3 \pm 1.2$ | $4.7 \pm 0.1$ | $6.2 \pm 1.6$ |
| 2b | $15.0 \pm 1.0$ | $30.2 \pm 2.9$ | $13.0 \pm 0.6$ | $8.2 \pm 0.4$ | $5.4 \pm 0.6$ |
| 3 | $7.6 \pm 1.3$ | $11.1 \pm 1.5$ | $6.5 \pm 0.4$ | $8.5 \pm 1.1$ | $9.5 \pm 1.0$ |
| 4 | >100 | >100 | >100 | >100 | >100 |
| 5 | >100 | $91.8 \pm 9.4$ | $7.7 \pm 1.1$ | $7.4 \pm 2.0$ | $3.0 \pm 0.3$ |

### 3.2. Effect of Selected Ferrocenes on Cell Cycle

To determine the effect of the tested ferrocenes on the cell cycle, PI assay was used. HeLa cells showed significant changes in cell cycle distribution, especially an increased proportion of cells in S phase and a corresponding decrease in G0/G1 phases (Figure 2). On the other hand, SiHa cells showed no significant changes in cell cycle distribution in response to the tested ferrocenes (data not shown). Importantly, the applied doses of irradiation in our experiment had no significant effect on cell cycle distribution except for increased accumulation of HeLa cells in the G2/M phase in response to **3**.

### 3.3. Analysis of Cell Death

Since MTT tests clearly confirmed the cytotoxicity of the selected compounds, closer examination of the specific mechanisms and pathways associated with cell death was performed.

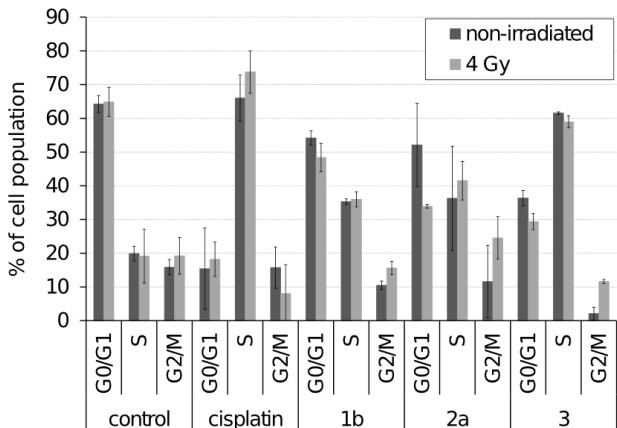

**Figure 2.** Cell cycle analysis of HeLa cells exposed for 24 h to 5 μM ferrocenes either with or without 4 Gy irradiation.

### 3.3.1. Effect of Selected Ferrocenes on ROS Production

A cell-membrane-permeable chloromethyl derivative of $H_2DCFDA$ was used to determine the production of reactive oxygen species. An increased amount of ROS was measured after 2 h of treatment with selected ferrocenes in 10 μM concentration. Untreated cells and cells treated with hydrogen peroxide were used as controls. The relative fluorescence values are summarized in Figure 3.

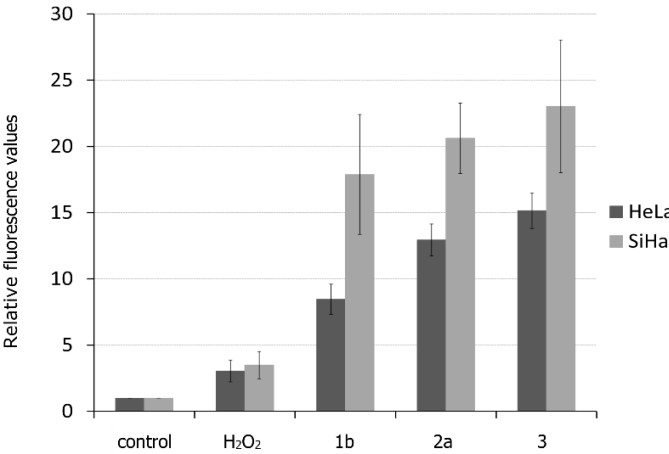

**Figure 3.** Reactive oxygen species (ROS) production in HeLa and SiHa cells after treatment with the most active ferrocenes. Untreated cells were used as a negative control, $H_2O_2$ was used as a positive control. Data were normalized to control untreated cells.

### 3.3.2. Effect of **1b**, **2a**, and **3** on Mitochondrial Membrane Potential

Increased levels of reactive oxygen species can be closely associated with mitochondrial metabolism [31–33]. Thus, the JC-1 probe, routinely used for measuring the state of mitochondrial membranes and their potential in a wide spectrum of cell types, was applied [34–37]. Increased mitochondrial potential in HeLa cells (by about 50% to 100%) was observed after treatment with 5 μM ferrocenes. SiHa cells were resistant under the same conditions, but increasing the concentration to 40 μM caused a similar change in mitochondrial potential (Figure 4).

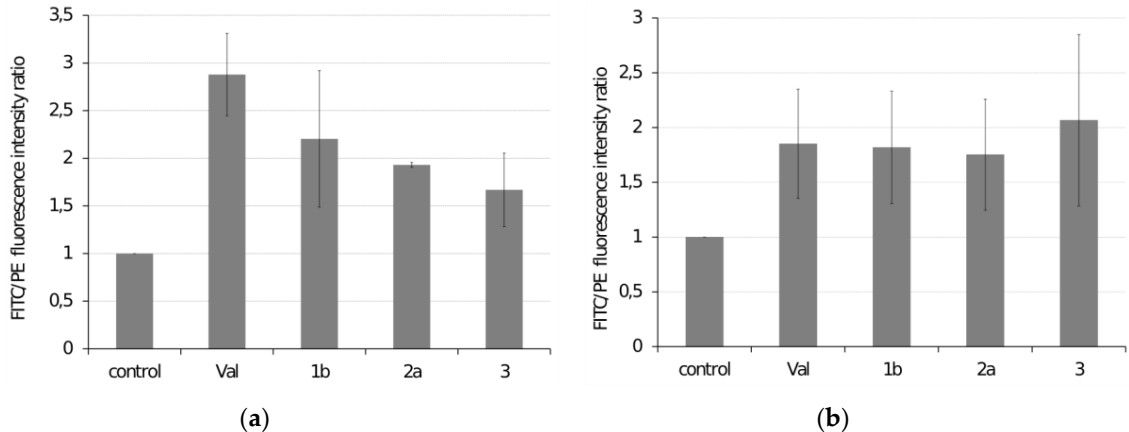

**Figure 4.** Determination of mitochondrial membrane depolarization in (**a**) HeLa cells exposed to 5 μM ferrocenes for 24 h and (**b**) SiHa cells exposed to 40 μM ferrocenes for 6 h. Valinomycin (Val) was used as a positive control. Fluorescein isothiocyanate (FITC) and Phycoerythrin (PE) fluorescence intensity ratio were determined. Data are reported in relation to untreated cells (control).

### 3.3.3. Analysis of Apoptosis

Increased production of reactive oxygen radicals along with the disruption of the mitochondrial membrane may indicate the involvement of apoptosis as a possible mode of cell death. Annexin V labeling with FITC was used for cytometric measurement of the translocation of phosphatidylserine to the outer surface of the plasma membrane [38–40]. PI staining of DNA was used to detect the late phase of apoptosis. Treatment with active ferrocenes was associated with the induction of both early and late phases of apoptosis in either cell line. Necrotic cells were not observed in significant amounts (Figure 5).

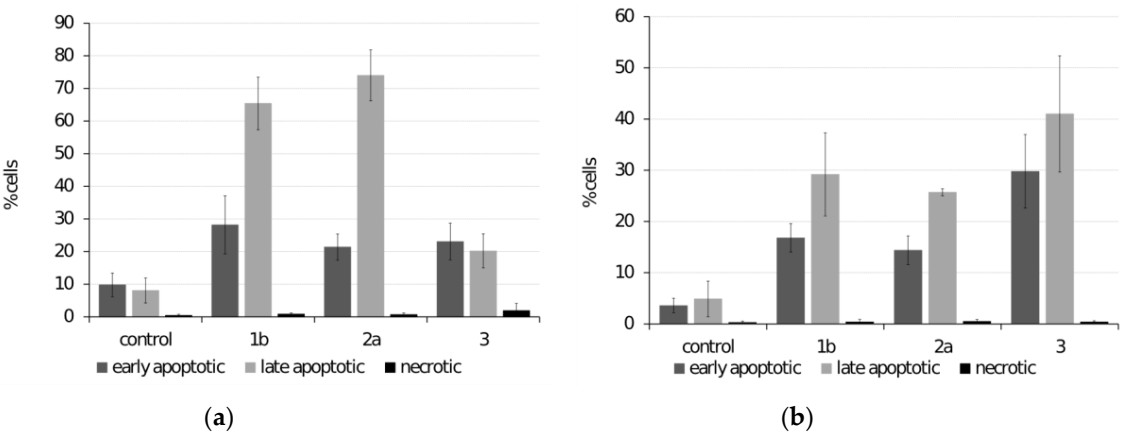

**Figure 5.** Determination of apoptosis by flow cytometry in (**a**) HeLa cells treated with 10 μM ferrocenes and (**b**) SiHa cells treated with 20 μM ferrocenes.

### 3.4. Autophagy Detection

The production of ROS is not specific only to apoptotic cells. Autophagy can also be induced by higher levels of reactive oxygen species [41,42]. To elucidate whether autophagy is also elevated in response to treatment with ferrocenes, the levels of autophagy-associated proteins p62 and cleaved LC3B were determined by immunofluorescent staining in HeLa cells. Indeed, increased levels of both p62 and LC3B were observed in these cells exposed to particular ferrocenes (Figure 6). Clear elevation of the LC3B level was also confirmed by Western blot analysis in both cell lines exposed to particular ferrocenes (Figure 7). In parallel, we also analyzed the combined effect of irradiation on LC3B cleavage. Interestingly, a clear decrease in the LC3B level was observed, predominantly in SiHa cells.

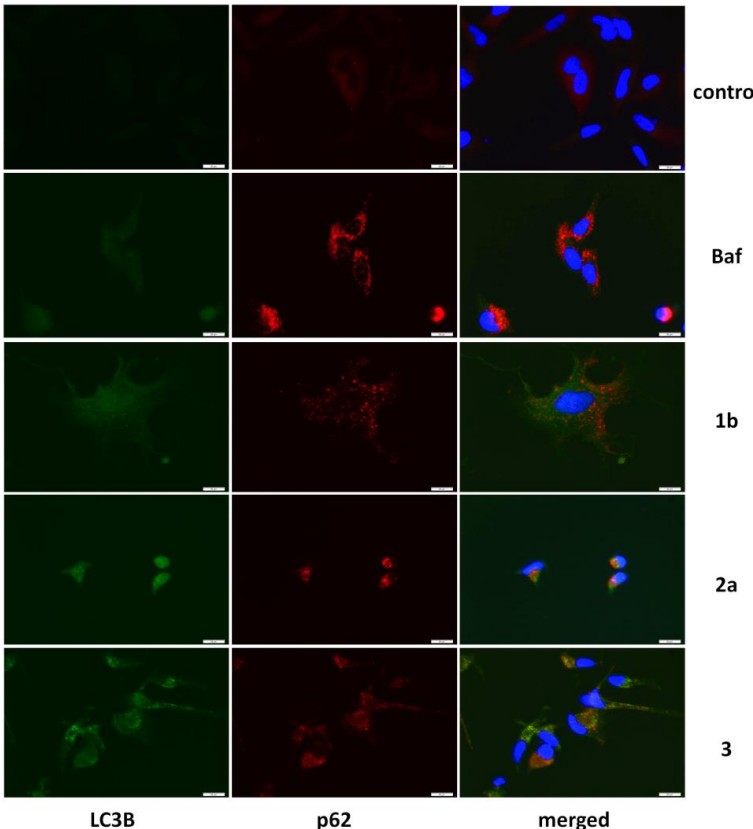

**Figure 6.** Detection of autophagy-related proteins in HeLa cells exposed to 5 µM ferrocenes for 24 h. Disruptor of autophagic flux Bafilomycin A (Baf) [43] was used as a positive control. Nuclei are stained with DAPI.

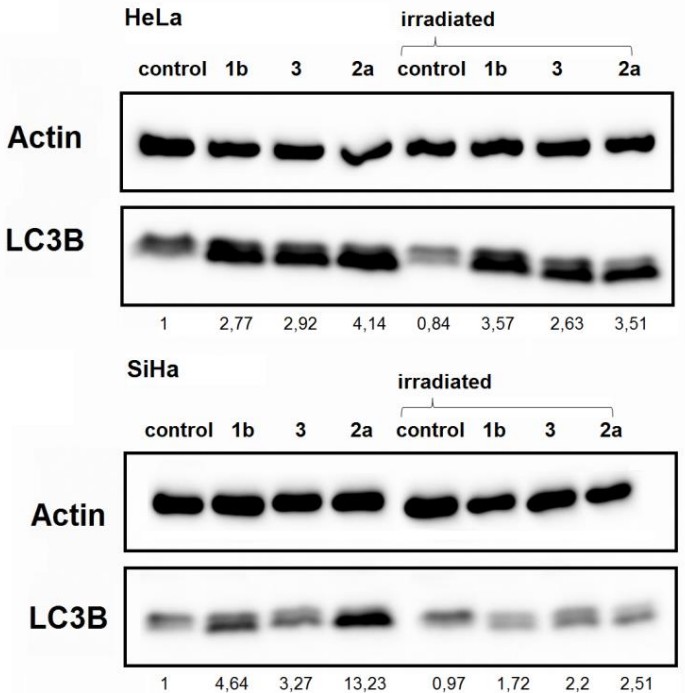

**Figure 7.** Detection of autophagy-associated protein LC3B in HeLa (upper part) and SiHa (lower part) cells. β-actin served as a loading control. Cells were treated with 2 µM concentrations of selected ferrocenes with or without a 4 Gy dose of irradiation.

### *3.5. Sensitivity to Ionizing Radiation*

Radiotherapy is a standard treatment option in advanced cervical carcinomas. Colony forming assay was used to evaluate the potential radiosensitizing effect of the selected ferrocenes [30].

### 3.5.1. Determination of the Surviving Fraction

The surviving fraction (SF) calculated from CFA is the ratio of surviving colonies in the irradiated and nonirradiated plates. Figure 8 shows that ferrocene **1b** had a potent radiosensitizing effect on both cell lines. In the SiHa cell line, a similar, albeit smaller, effect was observed for **2a**.

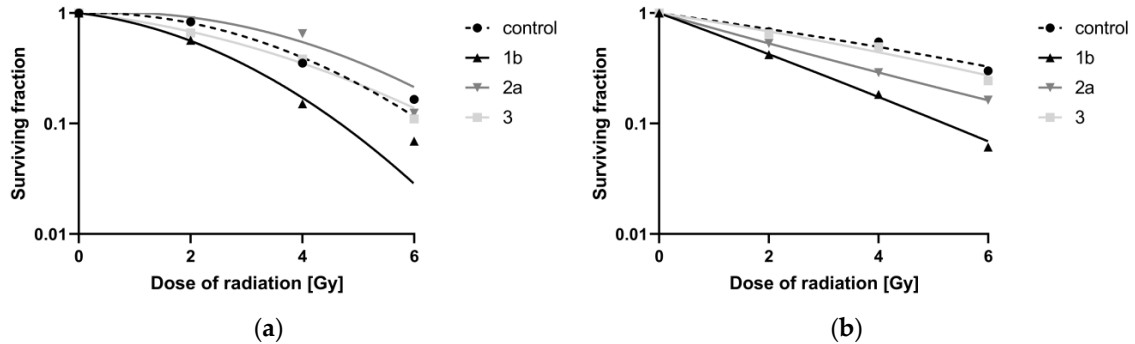

**Figure 8.** Graphical interpretation of the surviving fraction (SF) for the combination of selected ferrocenes with different doses of irradiation in (**a**) HeLa cells exposed to 1 μM ferrocenes and (**b**) SiHa cells treated with 2 μM ferrocenes.

### 3.5.2. Evaluation of Dose-Modifying Factors

Dose-modifying factors (DMFs) assess the additive effect of a tested drug when used in combination with ionizing radiation. They are the ratios of doses needed to obtain the same surviving fractions in cells irradiated with and without the tested ferrocenes [44]. Values below 0.8 were considered as showing an antagonistic effect, values between 0.8 and 1.2 as showing no effect, and values above 1.2 as showing a synergistic effect (Table 2).

**Table 2.** Dose-modifying factors for the selected ferrocenes in HeLa and SiHa cells.

| HeLa Cells | | | | SiHa Cells | | | |
|---|---|---|---|---|---|---|---|
| Conc. | 1b | 2a | 3 | Conc. | 1b | 2a | 3 |
| 0.5 μM | 0.9 ± 0.08 | 1.0 ± 0.1 | 0.9 ± 0.08 | 1 μM | 1.4 ± 0.19 | 1.3 ± 0.07 | 0.9 ± 0.08 |
| 1 μM | 1.6 ± 0.13 | 0.9 ± 0.2 | 1.2 ± 0.08 | 2 μM | 1.8 ± 0.5 | 1.4 ± 0.17 | 1.2 ± 0.08 |

## 4. Discussion

In recent articles, syntheses of some ferrocenes have been published [23,24]. Several of these compounds showed a clear cytotoxic effect that was equal or even greater when compared with cisplatin, which has become a mainstay of cancer therapy. We hypothesize that ferrocenes, similarly to cisplatin, may, in combination with ionizing radiation, show a synergistic cytotoxic effect. This radiosensitization effect could be particularly beneficial in the treatment of chemoresistant malignancies, e.g., cervical tumors that are generally perceived as resistant to cisplatin [45,46].

In previous studies, the mechanisms of transport of substituted ferrocenes were investigated using differential pulse voltammetry and inductively coupled plasma mass spectrometry. We identified membrane transferrin-receptor-mediated endocytosis of transferrin-bound ferrocene as the major mechanism of cellular uptake. Importantly, the rate of ferrocene accumulation in cancer cells is proportional to its cytotoxic effects [24,47]. To assess the exact mode of action of the studied ferrocenes, several methods were used. First, while no changes in cell cycle distribution were observed in SiHa cells, HeLa cells showed increased accumulation of cells in the S phase in response to treatment with all tested ferrocenes.

However, the increase in proportion of cells in the S phase predominantly after treatment with **1b** and **2a** was significantly lower when compared to that in cisplatin-treated cells. This could be of potential interest, since the doses of cisplatin used probably slow down DNA synthesis due to repairing its damage, which is associated with the accumulation of cells in the S phase [48], and cells in S phase are generally considered to be more radioresistant [49]. Thus, even though cisplatin is commonly used with radiotherapy, the effect of ionizing radiation in this combination is probably partly attenuated because cisplatin actually increases the proportion of cells in the S phase [50,51]. Therefore, treatment with ferrocenes resulting in the induction of cell cycle arrest in the S phase to a much lesser extent compared to cisplatin would be more effective, bringing greater benefit to these patients.

Second, a potent increase in reactive oxygen species was observed upon treatment with the selected ferrocenes. This can be explained by the basic chemical structure of the ferrocene core that is known as a catalyst in a Fenton reaction generating both hydroxyl radicals and higher oxidation states of the iron. Thus, in the presence of transition metals, $H_2O_2$, a product of mitochondrial oxidative respiration, is reduced inside the cells; this generates oxygen radicals responsible for damage to all macromolecules, including DNA, proteins, and membrane phospholipids that are damaged by the peroxidation of unsaturated fatty acids in exposed cells [52–54]. In recent years, several studies have confirmed potent induction of ROS by a range of organometallic complexes, including ferrocenes, in cancer cells [55–62]. These studies imply their possible utilization in cancer research and treatment. Our current results show a large increase in levels of reactive oxygen species after treatment with ferrocenes **1b**, **2a**, and **3**, which indicates their impact on the redox homeostasis in tumor cells.

Third, due to the aforementioned increase in ROS production, the effect of the tested compounds on mitochondrial function was also analyzed. Our experiments showed clear changes in mitochondrial membrane potential. These changes are generally considered to be an early step in apoptosis after treatment with different drugs [63,64]. Depolarization of the mitochondrial membrane leads to the release of cytochrome c from the mitochondria. This process triggers the formation of an apoptosome and the activation of caspases, launching the intrinsic apoptotic pathway [65]. Furthermore, early and late stages of apoptosis were both detected via Annexin V assay and DNA labeling by PI. These findings are in agreement with those of other studies showing that many different ferrocene-containing compounds, e.g., ferrocifens, Pt–ferrocene complexes [66,67], and ferrocenes combined with retinoids, are able to activate apoptotic cell death in cancer cells [68–70].

Elevated levels of ROS can also lead to the induction of autophagy as a defense mechanism that allows cell survival during stress conditions. An increase in autophagy-related proteins was observed, especially for cleaved-form LC3B. These changes were most prominent after treatment with ferrocenes **1b** and **2a**. It is well known that autophagy can play a dual role in response to drug treatment. On one hand, upregulation of autophagy could help in the formation of cancer cells resistant to chemotherapy [71]. On the other hand, excessive elevation of autophagy may assist in the induction of programmed cell death [72]. In our study, combination of ferrocenes with irradiation led to a decrease in the amount of LC3B protein. The decrease was most pronounced in response to ferrocene **2a**. It can therefore be assumed that autophagy induction helps cells to overcome exposure to ferrocenes. In line with these findings, combined treatment with ionizing radiation attenuates autophagy induction and results in significantly higher anti-tumor efficacy. This correlates with the CFA results, which showed decreased colony formation rates after combination of ferrocene **1b** and partially **2a** with radiotherapy when compared to radiotherapy alone. To quantify this interplay, the dose-modifying factor for each ferrocene was calculated. The most pronounced synergistic effect was observed for ferrocene **1b** (1.6× increased effectiveness of radiotherapy in HeLa and 1.8× in SiHa cell lines) and for ferrocene **2a** (1.4× for the SiHa cell line).

## 5. Conclusions

A series of ferrocenes based on the general formula [Fe($\eta^5$-C$_5$H$_4$CH$_2$(*p*-C$_6$H$_4$)CH$_2$(N-het))$_2$] bearing either substituted or unsubstituted saturated five- and six-membered nitrogen-containing heterocycles

showed higher cytotoxic activity than cisplatin against cervical cancer cell lines. These ferrocenes were able to increase the production of reactive oxygen species and disrupt mitochondrial homeostasis. As a consequence of these processes, the onset of a cell death mechanism via apoptosis and the activation of autophagy were observed. Furthermore, two of these ferrocenes—**1b** and **2a**—showed increased cytotoxic effects when combined with irradiation. However, the precise relationship between ionizing radiation and ferrocene derivatives needs further investigation. On the other hand, to put this problem into perspective, precise knowledge of the interaction between ionizing radiation and cisplatin is also unknown, and this holds true for many other chemotherapeutic drugs as well.

The combination of ferrocene **1b** and, to a lesser degree, **2a** with irradiation resulted in greater efficacy than either treatment alone in cervical cancer cell lines. Thus, our results suggest that the selected ferrocenes could be used in combination with radiotherapy, representing promising candidates for further investigation. Alternatively, the addition of ferrocene compounds to standard chemoradiation with cisplatin could also be considered. However, care needs to be taken to evaluate the potential increase in toxicity, since the studied ferrocenes and cisplatin have partly overlapping effects, mainly in the production of reactive oxygen species.

**Author Contributions:** Conceptualization, H.S. and V.R.; methodology, H.S.; software, H.S. and V.R.; validation H.S. and V.R.; chemical synthesis, J.P. and J.K., formal analysis, H.S.; investigation, H.S.; writing—original draft preparation, H.S.; writing—review and editing, V.R. and H.R.; visualization, H.S.; supervision, R.H.; project administration, R.H.; funding acquisition, J.P., J.K. and R.H. All authors have read and agreed to the published version of the manuscript.

**Funding:** This research was funded by GACR 17-05838S, MEYS-NPS I-LO1413 and MH CZ-DRO (MMCI, 00209805).

**Acknowledgments:** At this point we would like to thank L. Sommerova for help with flow cytometry analyses.

**Conflicts of Interest:** The authors declare no conflict of interest.

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
