# Peer review of "The Cytotoxic Effect of Newly Synthesized Ferrocenes against Cervical Carcinoma Cells Alone and in Combination with Radiotherapy"

_applsci, doi:10.3390/app10113728_

Round 1

Reviewer 1 Report

Skoupilova et al report that several ferrocenes exhibit
significant cytotoxicity against cervical cancer cells associated with increasing ROS production. Furthermore, the ferrocenes – 1b and 
2a show increased cytotoxic effect when combined with irradiation.

The manuscript is well written and the reported results are interesting. However, the characterization of the compounds is important. Please I suggest to report the HNMRs  (1 H1 and 13C) of compound 1c and previously published compounds.

It suggests to consult and to cite the following manuscripts:

-Sirignano et al. Synthesis, characterization and cytotoxic activity on breast cancer cells of new half-titanocene derivatives. Bioorganic & Medicinal Chemistry Letters 2013, 23, 3458–3462

-Saturnino et al. New Titanocene derivatives with high antiproliferative activity against breast cancer cells. Bioorganic & Medicinal Chemistry Letters, 24, 2014, 136–140

-Iacopetta et al. Novel Gold and Silver Carbene Complexes Exert Antitumor Effects Triggering the Reactive Oxygen Species Dependent Intrinsic Apoptotic Pathway. ChemMedChem Volume 12, Issue 24, December 19, 2017, 2054–2065 

-Sijongesonke Peter  et al. Ferrocene-Based Compounds With Antimalaria/Anticancer Activity Molecules, 2019, 24 (19) 3604

-Santos M et al. Recent Advances of Metallocenes for Medicinal Chemistry. Mini Rev Med Chem 2017, 17 (9), 771-784

-Staveren D et al. Bioorganometallic Chemistry of Ferrocene. Chem Rev 2004 Dec;104(12):5931-85

- Jaouen G et al. Ferrocifen Type Anti Cancer Drugs. Chem Soc Rev2015 Dec 21;44(24):8802-17

-Saturnino et al. N-heterocyclic carbene complexes of silver and gold as novel tools against breast cancer progression. Future Med. Chem.  2016, 8 (18), 2213-2229

Revision of the English language is recommended.

Author Response

We are delighted by valuable referee comments and suggestions on our manuscript.

Skoupilova et al report that several ferrocenes exhibit significant cytotoxicity against cervical cancer cells associated with increasing ROS production. Furthermore, the ferrocenes – 1b and 2a show increased cytotoxic effect when combined with irradiation. The manuscript is well written and the reported results are interesting. However, the characterization of the compounds is important. Please I suggest to report the HNMRs  (1 H1 and 13C) of compound 1c and previously published compounds.

Our response: Compound 1c is species consisting paramagnetic ferrocenium core. Therefore, NMR spectra showed very broad signals giving no information about structure. NMR spectra (together with other spectroscopic characterization) of remaining ferrocene derivatives were published previously (refs 23 and 24).

It suggests to consult and to cite the following manuscripts:

-Sirignano et al. Synthesis, characterization and cytotoxic activity on breast cancer cells of new half-titanocene derivatives. Bioorganic & Medicinal Chemistry Letters 2013, 23, 3458–3462

-Saturnino et al. New Titanocene derivatives with high antiproliferative activity against breast cancer cells. Bioorganic & Medicinal Chemistry Letters, 24, 2014, 136–140

-Iacopetta et al. Novel Gold and Silver Carbene Complexes Exert Antitumor Effects Triggering the Reactive Oxygen Species Dependent Intrinsic Apoptotic Pathway. ChemMedChem Volume 12, Issue 24, December 19, 2017, 2054–2065

-Sijongesonke Peter et al. Ferrocene-Based Compounds With Antimalaria/Anticancer Activity Molecules, 2019, 24 (19) 3604

-Santos M et al. Recent Advances of Metallocenes for Medicinal Chemistry. Mini Rev Med Chem 2017, 17 (9), 771-784

-Staveren D et al. Bioorganometallic Chemistry of Ferrocene. Chem Rev 2004 Dec;104(12):5931-85

- Jaouen G et al. Ferrocifen Type Anti Cancer Drugs. Chem Soc Rev. 2015 Dec 21;44(24):8802-17

-Saturnino et al. N-heterocyclic carbene complexes of silver and gold as novel tools against breast cancer progression. Future Med. Chem. 2016, 8 (18), 2213-2229

Our response: All proposed articles were included into article where appropriate.

Revision of the English language is recommended.

Our response: The manuscript was edited by native speaker to improve English.

Other changes:

Several typos have been corrected.

In abstract, we changed one sentence, since only compound 1c has been newly synthesized.

We also corrected 1c in Figure 1 and Scheme 1, since there should be Fe+ instead of Fe to get neutral chargé.

Reviewer 2 Report

The manuscript described anti-tumor activity of ferrocene derivatives based on combination therapy with radiotherapy or ferrocene derivative alone against cervical cancer. The authors revealed that developed ferrocene derivatives showed more potent than cisplatin in in vitro assay using cell lines derived from cervical cancer. Thus, evaluated ferrocene derivatives will be promising anti-tumor agents. Therefore, the manuscript is not too excellent to be published. In other words, the manuscript is so excellent that it should be published.

Comments

(1) Are counter ions such as BF4- safe to normal cells in vivo?

(2) Do ferrocene derivatives easily enter into cells across the membrane? And what are the permeation mechanisms?

(3) Will combination therapy with ferrocene derivatives, radiotherapy, and cisplatin effectively elicit anti-tumor activity?

(4) How was the signal transduction leading to ROS using ferrocene derivatives?

That is all.

Author Response

We are delighted by valuable referee comments and suggestions on our manuscript.

The manuscript described anti-tumor activity of ferrocene derivatives based on combination therapy with radiotherapy or ferrocene derivative alone against cervical cancer. The authors revealed that developed ferrocene derivatives showed more potent than cisplatin in in vitro assay using cell lines derived from cervical cancer. Thus, evaluated ferrocene derivatives will be promising anti-tumor agents. Therefore, the manuscript is not too excellent to be published. In other words, the manuscript is so excellent that it should be published.

  1. Are counter ions such as BF4- safe to normal cells in vivo? 
  2. Our response: BF4- anion is essentially non-toxic (LD50 > 5g/kg) as was found within in vivo experiments on rats (for more information see TOXICOLOGICAL EVALUATION No. 136 Tetrafluoroboric acid and its salts 11/2000, BG Chemie).
  3. Do ferrocene derivatives easily enter into cells across the membrane? And what are the permeation mechanisms? 
  4. Our response: In our previous work (Skoupilova et al., Eur J Pharmacol. 2020) we identified transferrin receptors as key mediators of intracellular accumulation of ferrocenes. Corresponding text including reference was added into discussion (lines 329-332).
  5. Will combination therapy with ferrocene derivatives, radiotherapy, and cisplatin effectively elicit anti-tumor activity? 
  6. Our response: Thank you for interesting comment. A potential combination of ferrocene derivatives, radiotherapy, and cisplatin is mentioned ion conclusion (lines 396-402).
  7. How was the signal transduction leading to ROS using ferrocene derivatives?
  8. Our response: Discussion was extended (lines 348-352)

Other changes:

Several typos have been corrected.

In abstract, we changed one sentence, since only compound 1c has been newly synthesized.

We also corrected 1c in Figure 1 and Scheme 1, since there should be Fe+ instead of Fe to get neutral chargé.